# An Urban Flood Model Development Coupling the 1D and 2D Model with Fixed-Time Synchronization

**Sang-Bo Sim and Hyung-Jun Kim ***

Department of Hydro Science and Engineering Research, Korea Institute of Civil Engineering and Building Technology, Goyang 10223, Republic of Korea; sangbo@kict.re.kr
* Correspondence: john0705@kict.re.kr; Tel.: +82-032-510-0276

**Abstract:** Due to climate change, the frequency and intensity of torrential rainfall in urban areas are increasing, leading to more frequent flood damage. Consequently, there is a need for a rapid and accurate analysis of urban flood response capabilities. The dual-drainage model has been widely used for accurate flood analysis, with minimum time step synchronization being commonly adopted. However, this method has limitations in terms of speed. This study applied the hyper-connected solution for an urban flood (HC-SURF) model with fixed-time step flow synchronization, validated its accuracy using laboratory observation data, and tested its effectiveness in real urban watersheds with various synchronization times. Excellent performance was achieved in simulating real phenomena. In actual urban watersheds, as the synchronization time increased, the errors in surcharge and discharge also increased due to the inability to accurately reflect water level changes within the synchronization time; however, overall, they remained minimal. Therefore, the HC-SURF model is demonstrated as a useful tool for urban flood management that can be used to advantage in real-time flood forecasting and decision-making.

**Keywords:** urban flood model; fixed-time synchronization; dual-drainage model

## 1. Introduction

Climate change is a serious global issue leading to more extreme weather patterns and frequent severe events such as urban floods [1]. Urban areas characterized by high population densities and extensive development are particularly vulnerable to these changes [2–4]. Torrential rainfall frequency in cities is increasing owing to climate change, often exceeding the capacity of storm sewer drainage systems. To mitigate urban flood damage, it is crucial to accurately assess urban watershed response capabilities and implement both structural and nonstructural alleviating measures [5]. Numerical models that accurately and rapidly calculate flood damage and reflect urban watershed characteristics are essential for evaluating urban flood response capabilities [6,7].

To simulate urban flooding numerically, models that analyze storm sewer flow and surface water dynamics are used. Traditional methods involve one-dimensional (1D) storm sewer network analyses to estimate flow, which is then used as input in surface water models to calculate the flood extent [8–10]. However, this approach can overestimate flood areas because it does not account for the reentry of surface water into the storm sewer network. This limitation can lead to inaccuracies in flood extent estimation, emphasizing the need for integrated models that consider the bidirectional flow between the surface and drainage systems.

Given the advances in computational power and data availability, the performance of models has improved in recent years from simple to sophisticated numerical models. Research has focused on dynamically linking storm sewer network flow models to surface water flow models of late. These integrated models can simulate the process by which storm sewers enter the drainage network, the flooding caused by flows exceeding the

network's capacity, and the reentry of residual surface water into the network, thereby providing a more accurate representation of urban flood dynamics.

Hsu et al. [11] combined storm sewer models, two-dimensional (2D) surface flow models, and pumping station operations to develop an urban flood model that was calibrated and validated for individual storms. Chen et al. [12,13] developed integrated models to simulate complex flow phenomena in urban drainage basins, and Leandro et al. [14] calibrated a 1D/1D urban flood model using the 1D/2D model results in the absence of field data. Djordjević et al. [15] created a new model for situations exceeding the hydraulic capacity of sewer systems, integrating it with a 2D surface flow model for urban flood simulation, achieving satisfactory matches with observed hydrographs and maximum surface flood levels. Dagnachew et al. [16] combined 1D and 2D flood models with dynamic bidirectional interactions based on water level differences between the sewer network and surface flow, tested in both virtual and real case studies. Fraga et al. [17] proposed a 1D–2D dual-drainage model for calculating rainfall–runoff transformations in urban environments. Recently, dual-drainage models utilizing the US Environmental Protection Agency's (EPA) open-source Storm Water Management Model (SWMM) have gained widespread use globally [18,19].

These studies laid the foundation for a more accurate interpretation of the physical characteristics of urban floods. However, the time step for 1D–2D flow exchange is generally much smaller than that for 1D models, which significantly reduces model efficiency [20,21]. Therefore, this study aimed to develop a hyper-connected solution for urban flood (HC-SURF) numerical model for urban flood prediction, which is validated through laboratory observations and real urban watershed flood analyses. By applying various rainfall scenarios, this study evaluated the accuracy of the HC-SURF model and proposed a method to efficiently reduce simulation time using fixed-time synchronization techniques. This approach aims to enhance the accuracy of urban flood predictions and assess the practicality of the HC-SURF model in simulating real phenomena.

## 2. Numerical Models

The HC-SURF model was developed to simultaneously interpret the flow of rainwater and surface water. The source code of SWMM 5.2 version [22], which is the most commonly used, was used for the sewer network flow analysis. The 2D surface water flow analysis model was developed directly, and the 2D shallow water equations were discretized using the finite volume method.

### 2.1. Two-Dimensional Surface Flow Model

The governing equation of the 2D model for surface water flow analysis is a 2D shallow equation, as shown in Equation (1).

$$\frac{\partial \mathrm{U}}{\partial t} + \frac{\partial \mathrm{F}}{\partial x} + \frac{\partial \mathrm{G}}{\partial y} = \mathrm{S} \tag{1}$$

$$\mathrm{U} = \begin{pmatrix} h \\ hu \\ hv \end{pmatrix}, \mathrm{F} = \begin{pmatrix} hu \\ hu^2 + \frac{gh^2}{2} \\ huv \end{pmatrix}, \mathrm{G} = \begin{pmatrix} hu \\ huv \\ hv^2 + \frac{gh^2}{2} \end{pmatrix},$$
$$\mathrm{S} = \begin{pmatrix} R - f \\ gh\left(S_{ox} - S_{fx}\right) \\ gh\left(S_{oy} - S_{fy}\right) \end{pmatrix} \tag{2}$$

In the system of Equation (2), U is the vector of the conserved variables, F and G are the flux vectors, and S is the source term vector. $u$ and $v$ are the velocity components in the $x$- and $y$-directions, respectively, and h is the water depth. $S_{ox}$ and $S_{oy}$ are the bottom slopes, and $S_{fx}$ and $S_{fy}$ are the bottom frictional terms along the $x$- and $y$-directions,

respectively. The governing equation is discretized using the finite volume method, as shown in Equation (3):

$$U_i^{n+1} = U_{ij}^n - \frac{\Delta t}{A}\left\{\sum_{k=1}^{N} F \cdot \Delta y - G \cdot \Delta x\right\} + \Delta t S_{ij}^n \tag{3}$$

Generally, the bottom frictional terms are expressed in terms of Manning's empirical formula, as shown in Equation (4) [23,24].

$$S_{fx} = \frac{n^2 u \sqrt{u^2 + v^2}}{h^{4/3}} \qquad S_{fx} = \frac{n^2 v \sqrt{u^2 + v^2}}{h^{4/3}} \tag{4}$$

where $n$ is Manning's relative roughness coefficient. In a diffusive wave approximation, the momentum equations do not consider local and advection accelerations. Thus, the simplified momentum Equation (5) can be described as follows:

$$\frac{\partial(h+z)}{\partial x} = \frac{\partial \eta}{\partial x} = \frac{n^2 u \sqrt{u^2 + v^2}}{h^{4/3}}, \qquad \frac{\partial(h+z)}{\partial y} = \frac{\partial \eta}{\partial y} = \frac{n^2 v \sqrt{u^2 + v^2}}{h^{4/3}} \tag{5}$$

By calculating the flow velocity using the above momentum equation, the following 2D flow velocity vector can be obtained, as shown in Equation (6):

$$V = -\frac{1}{n} h^{2/3} \sqrt{|\nabla \eta|} \frac{\nabla \eta}{|\nabla \eta|} \tag{6}$$

A depth-positivity-preserving condition technique was used to calculate the flow velocity at the boundary of the numerical grid. It is defined as the lower height of the boundary surface that is dominant in the flow analysis by comparing the lower heights of the left and right sides at the boundary surface of the grid. Subsequently, it was reset so that the water depth did not become a negative value compared to the water level on the left and right sides. By using the lower height of the boundary and the reset water depth, the water level conditions on the left and right sides of the boundary surface were reset, as shown in Equations (7) and (8), and applied to Equation (9). Next, the flow rate, $q = hu$, as shown in Equation (10), was calculated and substituted into the continuous equation to calculate the change in depth. The $\Delta t$ for updating the water level was calculated and applied after applying the CFL (Courant–Friedrichs–Lewy) condition.

$$z_b = max(z_L, z_R) \tag{7}$$

$$h_L = max(0, \eta_L - z_b), \ h_R = max(0, \eta_R - z_b), \ h_b = 0.5(h_L + h_R) \tag{8}$$

$$\hat{\eta}_L = h_L + z_b, \ \hat{\eta}_R = h_R + z_b \tag{9}$$

$$q_l = -sign(\hat{\eta}_R - \hat{\eta}_L)\frac{1}{n}h_b^{5/3}\sqrt{\left|\frac{\hat{\eta}_R - \hat{\eta}_L}{\Delta l}\right|} \tag{10}$$

### 2.2. One-Dimensional Drainage Network Model

In this study, SWMM 5.2 was adopted as the 1D dynamic sewer network model because it is the most widely used software for the hydrological and hydraulic modeling of urban catchments [22]. Moreover, the SWMM has a public source code; it is easy to couple with the surface water model. The hydrodynamics module of SWMM solves the 1D Saint–Venant equations for unsteady flow. These are referred to as dynamic wave analyses and are implemented in the extended transport (EXTRAN) module. The governing equations of SWMM are presented as Equations (11) and (12). The water depth in the manholes can

be calculated by solving the mass conservation equation. The momentum equation is used to calculate the velocity of the flows in the pipes.

$$\frac{\partial A}{\partial t} + \frac{\partial Q}{\partial x} = 0 \tag{11}$$

$$\frac{\partial Q}{\partial t} + \frac{\partial (Q^2/A)}{\partial x} + gA\frac{\partial A}{\partial x} + gAS_f = 0 \tag{12}$$

The EPA-SWMM is a 1D dynamic sewer network model developed for simulating water flow conveyance within drainage systems. The SWMM solves the 1D Saint–Venant equations for gradually varied, unsteady flow. The SWMM includes a dynamic link library that allows the retrieval and setting of hydraulic variables using other models during the simulation. It has recently been coupled with Iber to obtain the 1D/2D dual-drainage model Iber-SWMM [25].

Surface and sewer network equations were computed independently by each model. However, water exchange between the models occurs at every synchronization time step, ensuring a correct coupling and maintaining the mass balance of water. The interaction between the overland flow and sewer drainage system is limited to inlets and manholes. Surface water can enter the sewer network only through the inlets, whereas water can only return to the surface through manholes. Additionally, rainfall discharges on the roofs of buildings were computed using a subcatchment approach by solving a nonlinear reservoir equation [26,27].

*2.3. Coupling Method*

The HC-SURF model is a dual-drainage model in which the storm sewer network flow simulation and surface water flow simulation are conducted simultaneously. Flow synchronization was performed at fixed intervals according to the area of the analysis target region. By periodically sharing the calculated results, the inflow into the network or the outflow from the network to the surface can be estimated, allowing the urban flooding process to be simulated in a manner very similar to real phenomena.

The water levels in the manholes of the storm sewer network were compared with the surface water levels to calculate the exchange flow between the models. Depending on the water level conditions, either the orifice or weir formula was automatically selected to calculate the exchange flow. The flow exchange based on the water-level conditions was defined as follows: when surface water moves into a manhole, it is termed a discharge; when the flow reverses from the manhole to the surface, it is termed a surcharge. The discharge and surcharge can be calculated by comparing the elevation and water levels of the manholes and surface water. The exchange of flow between the surface water and the network was performed using the orifice and weir formulas given below [21].

If the manhole water level h was lower than the surface elevation $z$ (and, thus, lower than the surface water level), the discharge was calculated using the following weir formula:

$$q_{discharge} = c_w \times w_w \times h \times \sqrt{2 \times g \times h} \tag{13}$$

where $c_w$ represents the weir discharge coefficient, and $w_w$ represents the weir crest width. However, if the water level in the manhole rose above the surface elevation (but remained below the surface water level), that is, h > z, the drainage was calculated using the following orifice formula:

$$q_{discharge} = c_0 \times A_m \times \sqrt{2 \times g \times (h + z - H)} \tag{14}$$

Here, $c_0$ represents the orifice discharge coefficient and $A_m$ depicts the manhole surface area. Finally, if the water level in the manhole exceeded the surface water level, that is H > (h + z), the surcharge was calculated using the following orifice formula:

$$q_{surcharge} = c_0 \times A_m \times \sqrt{2 \times g \times (H - h - z)} \tag{15}$$

An analysis flowchart of the HC-SURF model is shown in Figure 1. The flow exchange between the surface and storm sewer networks was performed using a fixed-time synchronization technique. A detailed description of this process is provided in Section 4.3.

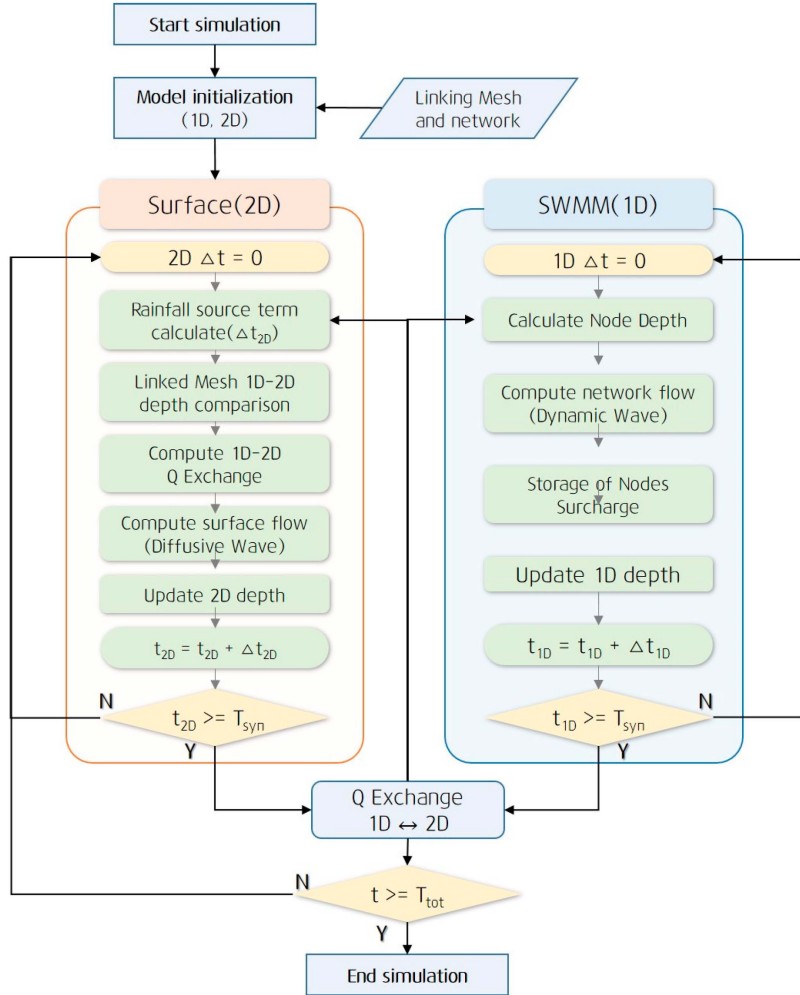

**Figure 1.** HC-SURF model flowchart.

### 3. Test and Validation

*3.1. Laboratory Experiment Setup*

To test the HC-SURF model, the model was applied to a mathematical experiment case for the exclusion of small urban areas. Sañudo et al. [27] studied how the spatial representation of roofs in urban drainage models might affect the model's results, based on the results of laboratory experiments. The facility represented a T-intersection street of 100 m² linked to a sewer system and was equipped with a rainfall simulator capable of generating spatially homogeneous rainfall intensities of 30, 50, and 80 mm/h. The sewer network (Figure 2) had a principal pipeline along the longitudinal dimension of the facility, consisting of four manholes connected by pipes with an inner diameter of 240 mm and a slope of 1%. Additionally, a transversal pipe with an inner diameter of 194 mm and a slope of 0.5% intersected the principal pipeline at manhole 3 (MH3). The XYZ coordinates, dimensions, and topology of the pipes, inlets, and nodes are listed in Table 1. The surface runoff enters the sewer system through four inlets of 0.5 × 0.2 m and a downstream transversal grate of 2.5 × 0.13 m that covers the roadway width (Table 1).

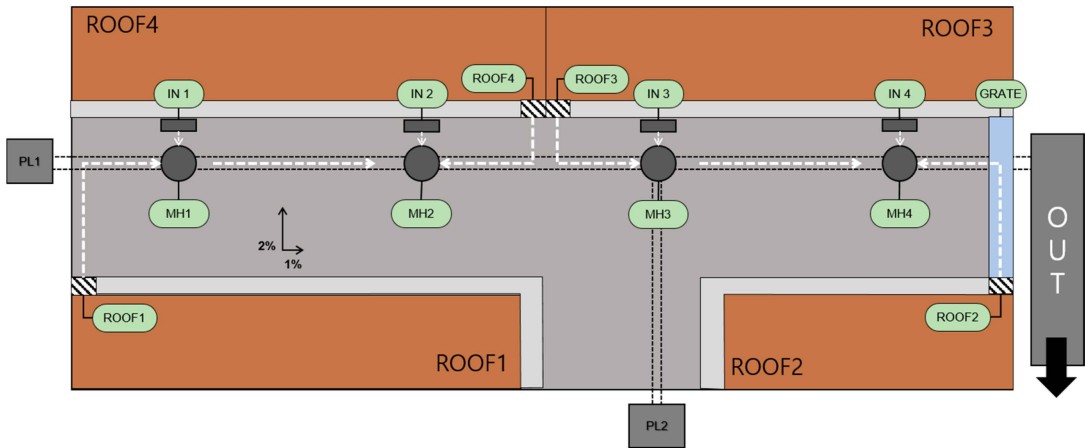

**Figure 2.** Laboratory schematic diagram.

**Table 1.** Laboratory specifications.

| Category | Identifier | Slope (%) | Length (m) | Width (m) | Manhole Connection | X Coord (m) | Y Coord (m) |
|---|---|---|---|---|---|---|---|
| ROOF | ROOF1 | 16 | 7.03 | 1.55 | MH1 | | |
| ROOF | ROOF2 | 26 | 4.59 | 1.55 | MH4 | | |
| ROOF | ROOF3 | 37 | 7.42 | 1.55 | MH3 | | |
| ROOF | ROOF4 | 51 | 7.54 | 1.55 | MH2 | | |
| INLET | INLET1 | | 0.5 | 0.2 | MH1 | −5.76 | 1.14 |
| INLET | INLET2 | | 0.5 | 0.2 | MH2 | −1.88 | 1.14 |
| INLET | INLET3 | | 0.5 | 0.2 | MH3 | 1.88 | 1.14 |
| INLET | INLET4 | | 0.5 | 0.2 | MH4 | 5.74 | 1.14 |
| INLET | GRATE | | 2.5 | 0.13 | MH4 | 7.37 | 0.005 |

### 3.2. Numerical Model Setup

All the areas were discretized using a triangular unstructured mesh with an average element size of 0.05 m. The roof gutter sections, responsible for collecting and transporting rainwater to the inlets, were modeled with a 0.07 m gutter section and a 0.03 m gutter wall section. The generated mesh consisted of 75,000 elements and 37,926 nodes (Figure 3). Manning's coefficients, referenced from previous studies, were set to n = 0.016 for street surfaces [28,29], 0.025 for roofs, and 0.008 for pipes. No initial conditions for the surface were defined because of pre-existing rainfall conditions in the laboratory data. The entire numerical model assumes impermeable areas, and the surface flow is influenced by the mesh geometry and Manning's coefficients. The rainfall map used was derived from rainfall characterization, introducing rasters with average rainfall intensities of 30.3, 54.2, and 85.0 mm/h, simplified to 30, 50, and 80 mm/h for the simulator.

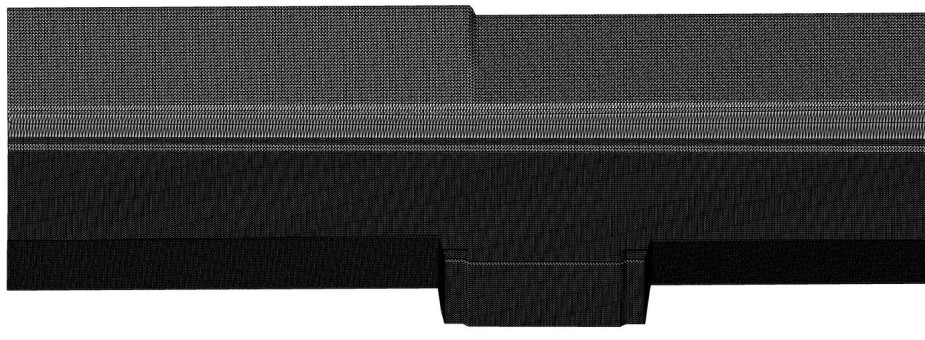

(**a**)

**Figure 3.** *Cont.*

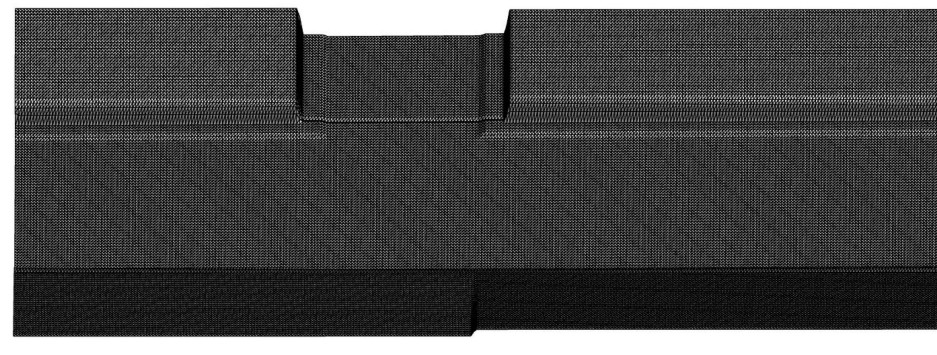

(**b**)

**Figure 3.** Construction of experimental model: (**a**) forward; (**b**) reverse.

*3.3. Test Results*

In this study, the accuracy of the HC-SURF model was validated through simulations under the T1, T2, and T3 rainfall scenarios. The rainfall scenarios of T1, T2, and T3 refer to rainfall events with a fixed rainfall intensity of 30, 50, and 80 mm/h for 4 min, respectively. The simulation results were compared with observed data to perform a mass balance analysis of the HC-SURF model based on different rainfall intensities. Additionally, the inflow capacities of various elements, such as manholes and roofs, were compared and analyzed to evaluate the model's suitability.

3.3.1. Mass Balance Check

To analyze the mass balance of the developed model, simulations were performed for different rainfall conditions, and the inflow capacity of each element was compared with the outflow capacity at the outlet (Figure 4). The difference between the total outflow from elements (roofs, gutters, manholes, and grids) and the outflow capacity at the outlet was 1.2 L/s for T1, 0.8 L/s for T2, and 0.3 L/s for T3. The mass balance error increased with lower rainfall, which was likely due to the lower flow velocity, causing an increase in residual surface water during the simulation period. Comparing the total flow rate at the outlet with the sum of the flow rates from each element, the differences were found to be less than 1% for rainfall scenarios T1, T2, and T3. This indicates that the developed model has appropriate flow control capability and can effectively manage the overall flow.

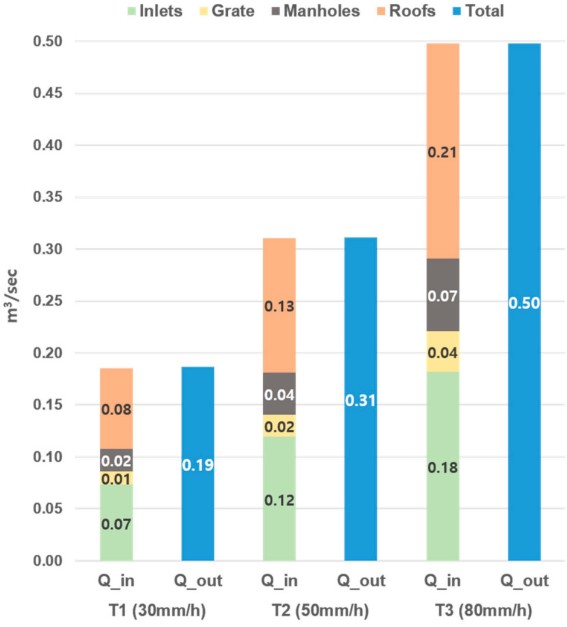

**Figure 4.** Mass balance analysis results.

### 3.3.2. Comparison of Inflow Capacities by Element

Inflow capacity analysis by element revealed that the roofs captured the most water, followed by the inlets, manholes, and outlets. Simulations of rainfall events T1–T3 showed no overflow from the manholes or inlets, indicating that all the flows were captured by the elements. A comparison with observational data showed an error of less than 1% in all cases, confirming the excellent reproducibility of the experimental results (Figure 5).

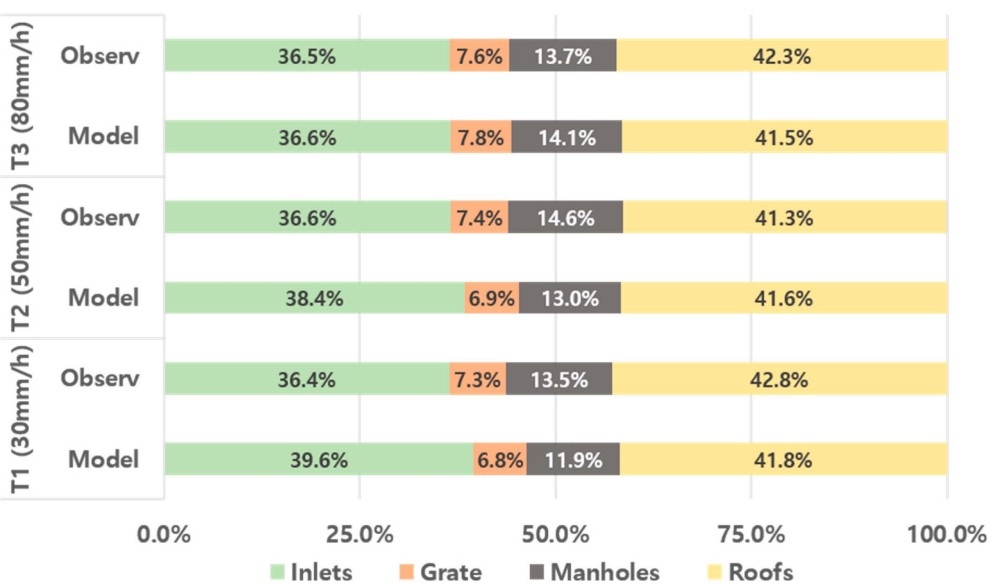

**Figure 5.** Analysis of inflow capacities by element.

Figure 6 shows the comparison of inflow rates for each element under rainfall events T1–T3. The inflow rates of the inlets were proportional to their respective catchment areas. Inlet 1, located at the uppermost part of the laboratory setup, had the smallest catchment area and the lowest inflow rate. Inlet 3, located at an intersection, had the largest catchment area and, consequently, the highest inflow rate.

The mean absolute error (MAE) and root mean square error (RMSE) were used to verify the accuracies of the calculated and observed hydrological curves. The MAE showed excellent accuracy with 0.0035–0.1267 L/s and RMSE of 0.0049–0.1772 L/s. As the rainfall intensity increased, the error tended to increase slightly, and some inlet values were somewhat different from the observed values. In particular, in the case of Inlet 4, you can check the delay increase and delay outflow. Various factors were involved, such as the location of the inlet during the numerical simulation, nonuniformity of the laboratory topography, and observation error.

However, these errors were offset by the inflow of other inlets and, consequently, it can be seen that the hydrological curve of the outfall agrees well. This was because the error by inlet was offset by other inlets and outfalls.

Figure 7 presents a comparison of the inflow capacities of the roof inlets. As the roofs progressed from ROOF1 to ROOF4, their slopes increased. It was expected that the inflow would increase and that the hydrograph would steepen accordingly. However, both numerical simulations and observed values showed minimal variation in inflow capacity with changes in roof slope. ROOF4 can see the delay rise and delay runoff. This, like the error in INLET4, is believed to be due to the shape difference between the actual roof and the grid created by the small grooves or protruding parts of the laboratory. During the T3 simulation, the difference in inflow capacity owing to the roof slope was approximately 1 L/s, with the highest inflow observed for ROOF3, which had the second-steepest slope.

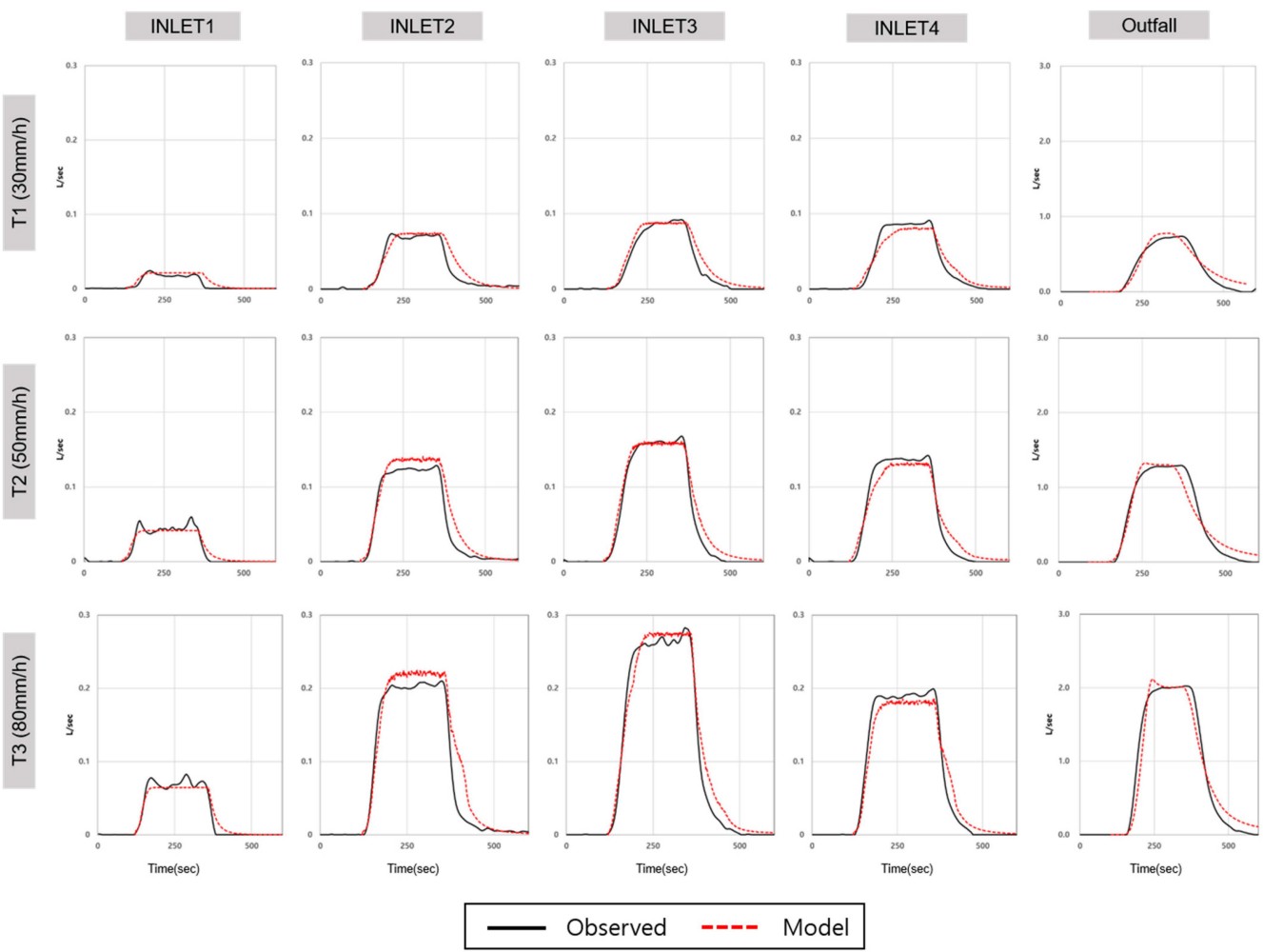

**Figure 6.** Comparison of observed and model inlet inflow capacities.

The accuracy of the roof inlet simulations was also validated using MAE and RMSE. The average MAE for all simulations was 0.0043 L/s, the RMSE was 0.0068 L/s, and the maximum error was 0.0047 L/s, indicating better agreement with the observed values than the ground inlets. This suggests that the actual shape of the roof was well represented in the numerical simulation.

For all elements, the average error was an MAE of 0.0157 L/s and an RMSE of 0.0217 L/s. The HC-SURF model exhibited high accuracy in matching the observed results across roofs, inlets, and outlets, demonstrating excellent performance in simulating real-world phenomena (Table 2).

**Table 2.** Quantitative review of inlet inflow capacities: observed vs. model results.

| Elements | T1 (30 mm/h) | | T2 (50 mm/h) | | T3 (80 mm/h) | |
|---|---|---|---|---|---|---|
| | MAE | RMSE | MAE | RMSE | MAE | RMSE |
| INLET_1 | 0.0035 | 0.0049 | 0.0038 | 0.0056 | 0.0055 | 0.0078 |
| INLET_2 | 0.0069 | 0.0103 | 0.0114 | 0.0153 | 0.0187 | 0.0273 |
| INLET_3 | 0.0074 | 0.0093 | 0.0082 | 0.0110 | 0.0144 | 0.0182 |
| INLET_4 | 0.0079 | 0.0091 | 0.0101 | 0.0118 | 0.0132 | 0.0165 |
| OUT | 0.0536 | 0.0672 | 0.0800 | 0.1142 | 0.1267 | 0.1772 |
| ROOF_1 | 0.0029 | 0.0048 | 0.0039 | 0.0057 | 0.0052 | 0.0100 |
| ROOF_2 | 0.0027 | 0.0044 | 0.0039 | 0.0055 | 0.0047 | 0.0081 |
| ROOF_3 | 0.0036 | 0.0045 | 0.0046 | 0.0069 | 0.0043 | 0.0069 |
| ROOF_4 | 0.0042 | 0.0064 | 0.0058 | 0.0100 | 0.0054 | 0.0082 |
| AVERAGE | 0.0103 | 0.0134 | 0.0146 | 0.0207 | 0.0220 | 0.0311 |

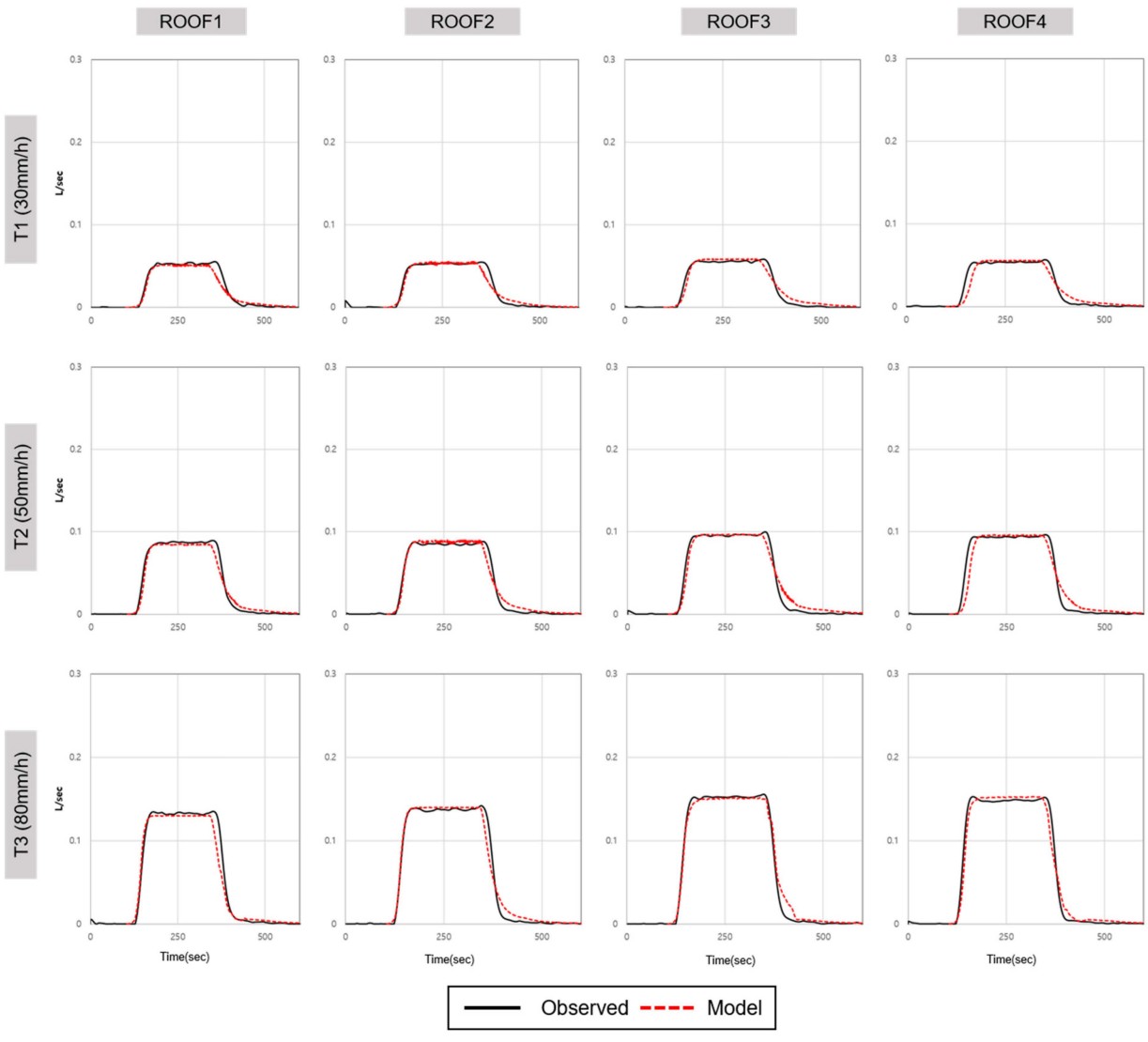

**Figure 7.** Comparison of observed and model roof inlet inflow capacities.

## 4. Urban Inundation Modeling

Record-breaking heavy rains occurred in Seoul, South Korea, on 8–11 August 2022. The southern part of Seoul in particular was subjected to cumulative daily precipitation of up to 381.5 mm and the instantaneous maximum precipitation was 141.5 mm/h, causing enormous human and property damage. The flooding phenomenon was reproduced by applying the HC-SURF model to this flood event, and the applicability of the proposed technique was reviewed.

### 4.1. Study Area

The study area is the Sillim drainage district in the Dorimcheon basin, which experienced flooding damage due to heavy rainfall in 2022. Dorimcheon is a local stream and secondary tributary of the Han River. The basin encompasses an area of 42.50 km$^2$ and a channel length of 14.51 km, with a dendritic pattern. The Sillim drainage district is located in the midstream area of the mainstream of Dorimcheon and has a drainage area of 5.14 km$^2$. Because of its lower elevation compared to adjacent drainage districts, surface water is expected to be concentrated in this low-lying area during rainfall events. The location of the Sillim drainage district and the positions of the rainfall observation stations are shown in Figure 8.

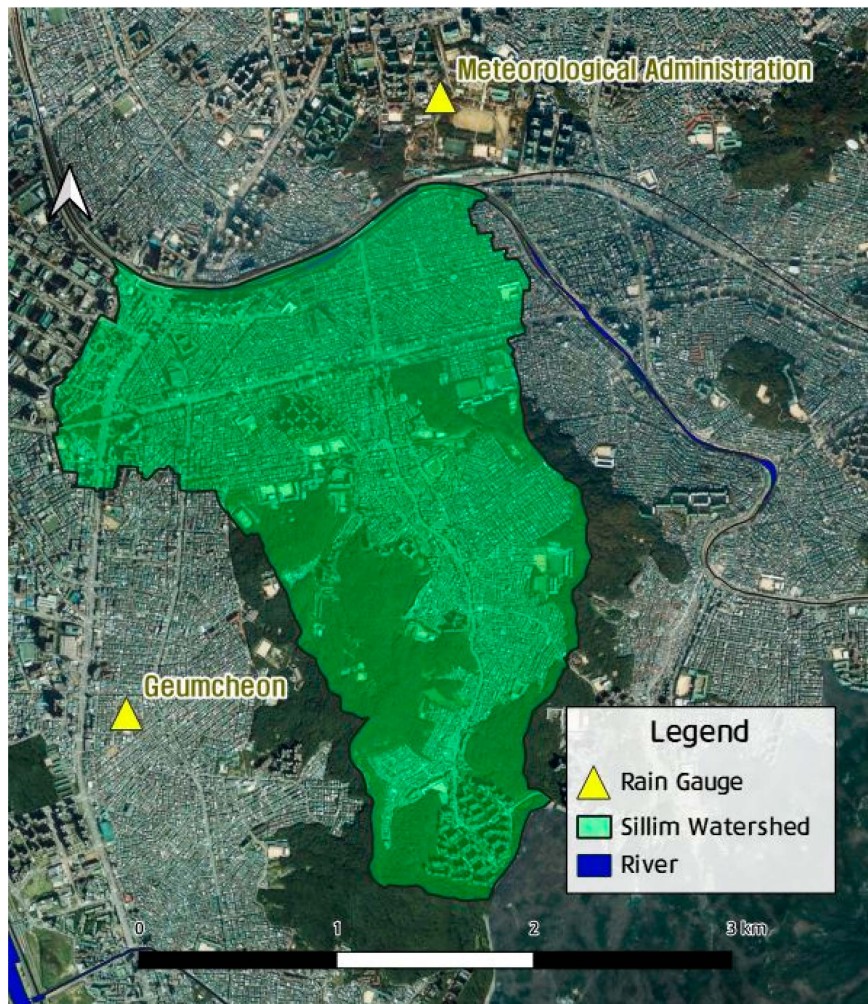

**Figure 8.** Status of the study area.

*4.2. Test Setup*

Hydrological and hydraulic data were established to perform a numerical analysis using the HC-SURF model. Hydraulic data were derived from the "Flood Control Plan Report for Specific River Basins" [30] to initialize the storm sewer drainage network parameters. The study area's sewer network consists of 3855 nodes, 4086 links, 18 pumps, and 21 channels. GIS data, including digital terrain maps, detailed soil maps, and land use data, were utilized to analyze hydrological factors such as the impervious area ratio, watershed slope, land cover, and infiltration capacity. The antecedent moisture condition II, representing typical soil moisture conditions with moderate runoff potential, was used. The land use status, detailed soil map, hydrological soil group, and digital elevation model analysis results for the Sillim drainage basin are shown in Figure 9.

The Sillim drainage district in the study area experienced significant human and property damage owing to the overflow of the Dorimcheon River caused by heavy rainfall August 2022. This study utilized that rainfall event to perform initial simulations and validate the model by comparing the 2022 Seoul flood trace map with the results of the 2D flood analysis. To accurately reflect the characteristics of the rainfall, minute-by-minute rainfall data were obtained from the Korea Meteorological Administration's (KMA) automatic weather system. The Thiessen weighting method was applied to calculate rainfall for the Sillim drainage district using rainfall data from nearby observation stations (KMA, Geumcheon, Gwanak, etc.). The rainfall events are listed in Table 3.

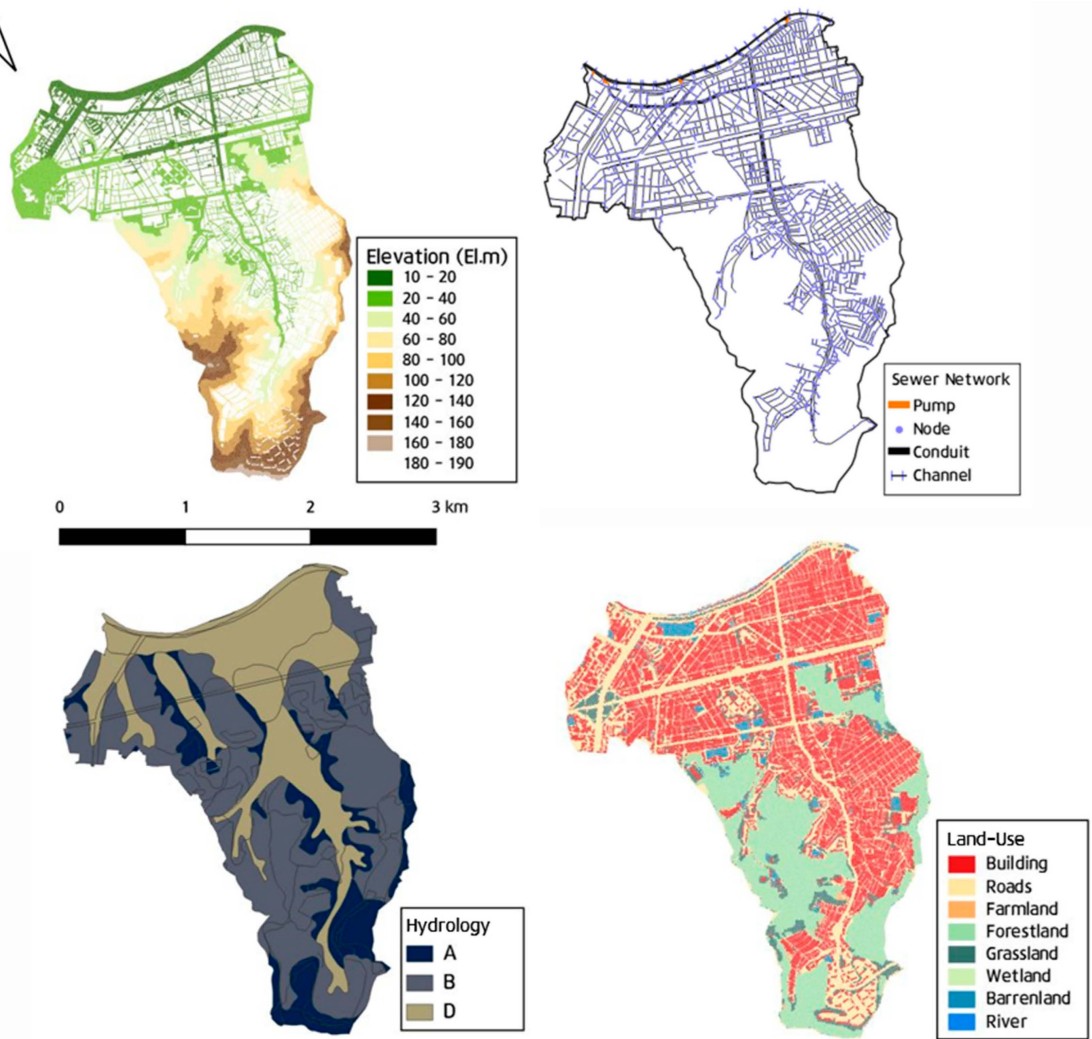

**Figure 9.** Hydraulic–hydrological data analysis and initial model setup.

**Table 3.** Rainfall scenarios.

| Station | Rainfall Amount (mm) | Rainfall Intensity (mm/h) | Thiessen Coefficient |
|---|---|---|---|
| Meterological Administration | 515 | 141.5 | 0.64 |
| Geumcheon | 445 | 94.0 | 0.36 |

The maximum hourly rainfall for the study area was recorded at 141.5 mm/h at the KMA rainfall observation station, exceeding the Sillim drainage district's storm sewer network design capacity of 95 mm/h (30-year return period). The maximum 24 h accumulated rainfall was observed to be 434.5 mm. To set the external water level boundary conditions, a 50-year return period was designed for the flood level (EL. 15.23 m) from the "Flood Control Plan Report for Specific River Basins." The simulation results, which involved exchanging the flow at minimum time steps (dt), were compared and analyzed against the flood trace map.

In Figure 10, the gray areas represent the flooded regions according to the inundation trace map, whereas the blue areas show the simulation results. In areas adjacent to Dorimcheon, flooding occurred because of the inability to drain internal water and the insufficient capacity of the storm sewer pipes, resulting in flooding in regions that closely matched the flood trace map. However, the flooding in low-lying areas was somewhat underestimated

in the simulation because the overflow from Dorimcheon was not considered. The primary cause of flooding in the analyzed rainfall event was the overflow of Dorimcheon, which affected the Sillim drainage district. Flood damage occurs because of overtopping caused by structural issues at water levels below the height of the river embankment.

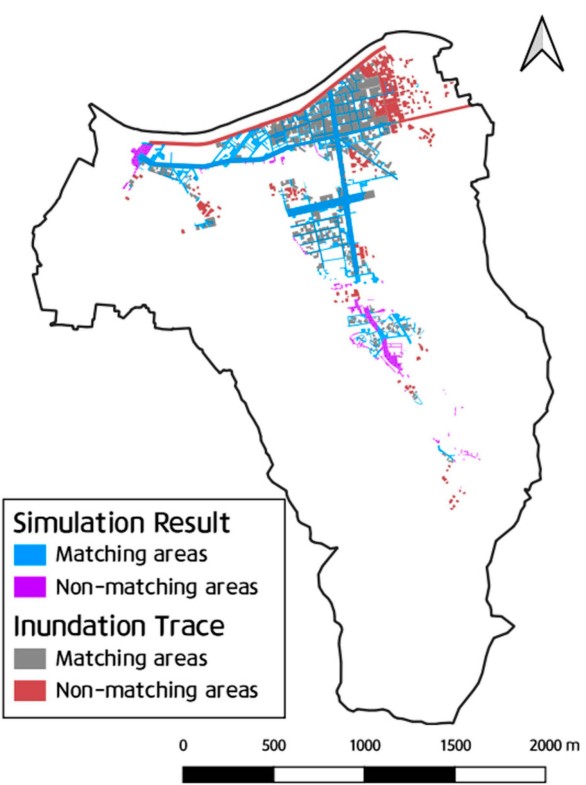

**Figure 10.** Comparison of analysis results and inundation trace map.

### 4.3. Fixed-Time Synchronization

There are two methods for synchronizing time in the dual-drainage model: using the minimum $\Delta t$ based on the CFL condition of the storm sewer network (1D) and surface water (2D), and using a predefined synchronization time. Synchronization based on the minimum $\Delta t$ is simple in terms of the model's synchronization process but is highly inefficient. If synchronization is based on the 2D $\Delta t$, which can be as small as 1 microsecond depending on the grid size, the computational time required becomes very large.

Therefore, this study applied a fixed-time synchronization technique. To achieve an effect similar to that of the minimum $\Delta t$ synchronization method, the following approach was used:

(1) At the user-defined synchronization time, store the node information for the 1D model, including the inflow, outflow, and water levels.

(2) Transfer the stored information to the 2D model. During this process, to account for variations within the fixed synchronization time, transfer the water levels from both the previous and the current exchange points and perform linear interpolation, as shown in Equation (16).

(3) Calculate the surface flow up to the synchronization time, then transfer the synchronized inflow, outflow, and water level information back to the 1D model. Repeat steps 1 to 3 throughout the simulation period to complete the fixed-time synchronization, as shown in Equations (17) and (18).

This approach balanced efficiency with accuracy, allowing the model to effectively simulate real-world phenomena while reducing computational time.

$$Depth^i_{1D} = \frac{Depth_{new} - Depth_{old}}{Syn} \cdot \Delta t_{2D} + Depth_{old} \tag{16}$$

where $Depth^i_{1D}$ represents the manhole water level at the $i$th synchronization time, $Depth_{new}$ represents the water level at the current synchronization time, and $Depth_{old}$ represents the water level at the previous synchronization time.

$$Volume_{2D(T_{sink}||T_{over})} = \sum_{n}^{n+syn} Q(x) \cdot \Delta t_{2D} \tag{17}$$

$$Volume^i_{1D(T_{sink}||T_{over})} = \frac{Volume_{2D(T_{sink}||T_{over})}}{syn} \cdot \Delta t_{1D} \tag{18}$$

Here, $Volume_{2D(T_{sink}||T_{over})}$ represents the total inflow and outflow volumes in the 2D model, and $Volume^i_{1D(T_{sink}||T_{over})}$ represents the ith inflow volume in the 1D model.

In this study, synchronization was performed at intervals of 10, 30, 60, 120, 300, and 600 s, respectively. The simulation results were analyzed to compare the differences in simulation outcomes based on synchronization times during the actual urban flood analysis.

## 5. Analysis Results

### 5.1. Analysis Results of Flow Exchange by Synchronization Time

The total inflow and outflow rates were compared to evaluate the simulation performance of the model for different synchronization times. Figure 11 presents a comparison of the manhole surcharge for synchronization times ranging from 2D $\Delta t$ to 10, 30, 60, 120, 180, 300, and 600 s, respectively. The total surcharge was similar across all synchronization times. However, an error of approximately 11,120.2 m$^3$ occurred at the 600 s synchronization time compared to the 2D $\Delta t$ simulation results, corresponding to 2.03% of the total surcharge amount. The reason for the minimal error in the outflow volume for different synchronization times is that the synchronization of the flow was based on the surface and manhole water levels at the respective times. Because the water-level variation within the synchronization time was minimal, it had little impact on the exchange flow calculations. However, when the synchronization time exceeds 300 s, it becomes challenging to reflect water-level changes in the manholes using linear interpolation, leading to errors.

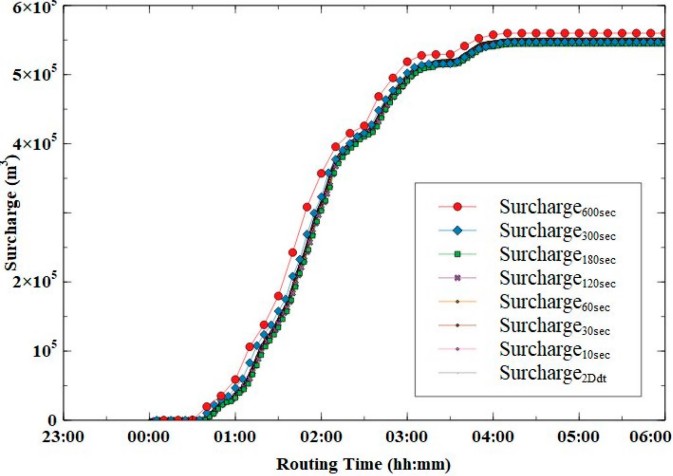

**Figure 11.** Cumulative surcharge analysis by synchronization time.

Figure 12 portrays the analysis of the discharge volume for different synchronization times. Although the overall discharge volume and timing of the peak discharge were similar, there was a tendency to underestimate the peak discharge as the synchronization time increased. This discrepancy became significantly more pronounced when the synchronization time exceeded 120 s, with an error of up to 9.9 m$^3$/s occurring at the 600 s synchronization time, compared to the 2D $\Delta t$ synchronization results. An error of approximately 10 m$^3$/s is not negligible, even when considering the large area of the study region, indicating the importance of selecting an appropriate synchronization time.

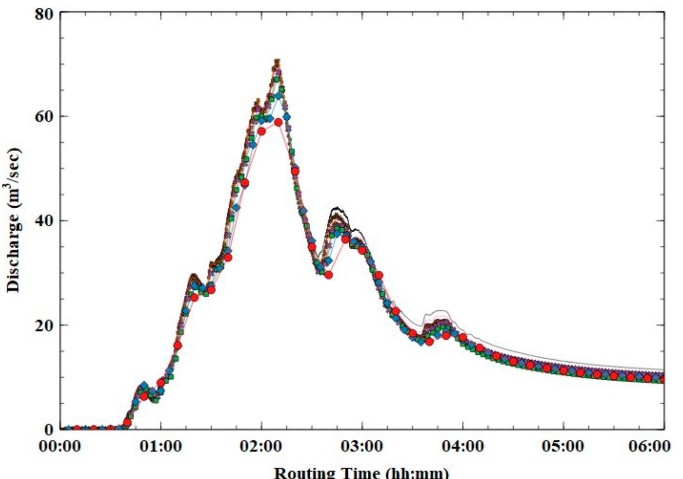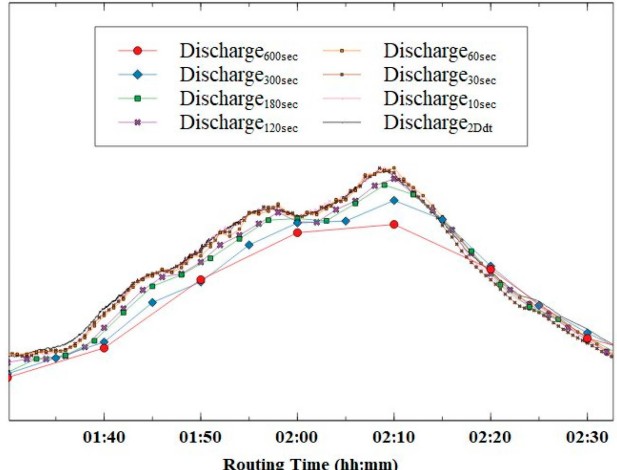

**Figure 12.** Analysis of discharge by synchronization time.

To quantitatively compare the surcharge and discharge for each synchronization time, analyses were performed using the MAE and RMSE metrics (Table 4). Both surcharge and discharge recorded increases in MAE and RMSE values as the synchronization time increased, with the discharge exhibiting a more significant increase.

**Table 4.** Quantitative comparison of surcharge and discharge for different synchronization times.

| Sync Time | Surcharge (m$^3$/s) | | Discharge (m$^3$/s) | |
|:---:|:---:|:---:|:---:|:---:|
| (s) | RMSE | MAE | RMSE | MAE |
| 10 | 2.00 | 0.94 | 0.96 | 0.78 |
| 30 | 2.01 | 0.95 | 1.65 | 1.38 |
| 60 | 2.08 | 1.04 | 1.59 | 1.33 |
| 120 | 2.08 | 1.02 | 1.80 | 1.49 |
| 180 | 2.33 | 1.01 | 2.36 | 2.07 |
| 300 | 2.17 | 1.14 | 2.80 | 2.19 |
| 600 | 2.39 | 1.42 | 3.70 | 2.72 |
| Average | 2.15 | 1.07 | 2.12 | 1.71 |

The MAE for surcharge increased from 0.94 m$^3$/s to 1.42 m$^3$/s, and the RMSE increased from 2.00 m$^3$/s to 2.39 m$^3$/s. Regarding discharge, the MAE increased from 0.78 m$^3$/s to 2.72 m$^3$/s, and the RMSE increased from 0.96 m$^3$/s to 3.70 m$^3$/s. These results confirm that flow exchange errors increase with longer synchronization times.

The reason for the increasing error with longer exchange periods is that the variation in surface water depth is greater and occurs over shorter intervals than during the exchange period, making it difficult to reflect these changes accurately using linear interpolation. The analysis area comprised a complex storm sewer network and building areas, and the maximum rainfall intensity of 141 mm/h exacerbated the error magnitude.

Figure 13 displays the analysis results of the maximum inundation area for different synchronization times. Although the inundation areas were similar across all synchroniza-

tion times, the maximum inundation area tended to increase as the synchronization time increased because of an increase in surcharge and a decrease in discharge.

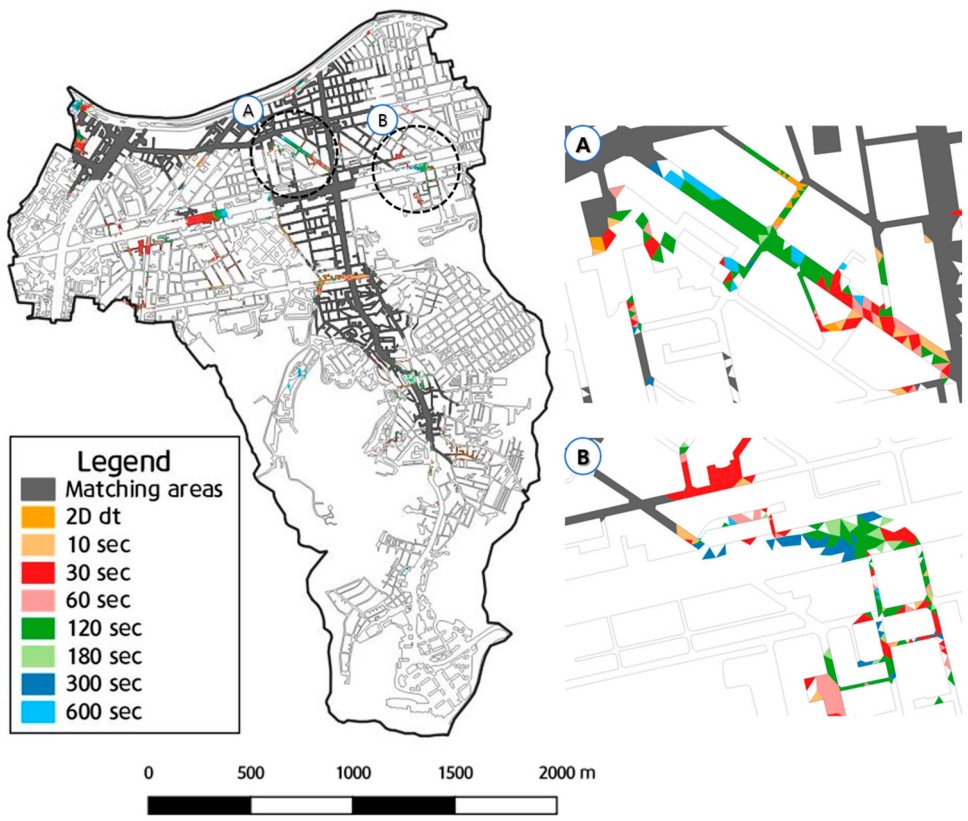

**Figure 13.** Comparison of flooded areas by synchronization time.

In areas A and B, the differences in the inundation area were pronounced, showing an overestimation of the inundation area compared to the 2D $\Delta t$-based synchronization. The difference in the maximum inundation area was overestimated by 14,855 m$^2$ for the 600 s synchronization time, corresponding to an error of approximately 4.7%. Given that the total area of the analysis region was 5.14 km$^2$ and the surface grid area, excluding building areas, was 3.06 km$^2$, the error in the inundation area owing to synchronization time differences was considered minor relative to the size of the analysis region.

*5.2. Analysis of Calculation Time by Synchronization Interval*

To evaluate the computational performance for different synchronization times, the calculation time of the method using 2D $\Delta t$-based synchronization was compared with the method using increased synchronization times. The comparison of calculation times was made by comparing the reduction rate of the simulation time relative to the 2D $\Delta t$-based synchronization. Therefore, the enhancement in computational speed (ECS) was calculated using Equation (13).

$$ECS = \frac{Time_{base} - Time_{Sync}}{Time_{base}} \tag{19}$$

Here, $Time_{base}$ refers to the calculation time for minimum time synchronization and $Time_{Sync}$ refers to the calculation time for fixed-time synchronization.

The analysis results indicate that the ECS ranged from 14.7% to 20.5%, with a tendency to increase as the synchronization time increased (Table 5). However, the ECS decreased slightly when the synchronization time exceeded 180 s, with the highest ECS observed at the 180 s synchronization time. This suggests that, as the synchronization time increases, the flow exchange between the storm sewer network and surface water becomes more limited, thereby reducing the time required for synchronization. However, additional

calculation time is required for the 2D and 1D computations. The complexity of the storm sewer network and the small grids between buildings in the study area may also have affected this.

**Table 5.** Analysis of calculation time by synchronization interval.

| Sync Time | Computational Time (s) | | | ECS (%) |
|---|---|---|---|---|
| | **Total** | **2D** | **1D** | |
| 2D dt | 1368.6 | 1390.2 | 1125.5 | |
| 10 s | 1167.2 | 1167.9 | 981.4 | 14.7 |
| 30 s | 1159.1 | 1161.0 | 954.8 | 15.3 |
| 60 s | 1112.3 | 1112.6 | 931.5 | 18.7 |
| 120 s | 1110.7 | 1110.9 | 941.7 | 18.8 |
| 180 s | 1088.3 | 1088.5 | 912.9 | 20.5 |
| 300 s | 1124.0 | 1124.1 | 931.9 | 17.9 |
| 600 s | 1093.4 | 1093.5 | 921.8 | 20.1 |

The variation in the computational efficiency with different synchronization times indicates that further research is required.

## 6. Discussion

The results from applying the HC-SURF model in this study reveal both its strengths and potential areas for improvement. The fixed-time synchronization technique greatly enhanced computational efficiency, particularly in large-scale urban watershed simulations, without significant loss of accuracy. This model, capable of handling 1D–2D flow exchanges in real time, shows potential for wide application in urban planning and disaster management. However, several factors require further discussion and research.

Sensitivity to Synchronization Time: This study found that computational efficiency improved as the synchronization time increased up to 180 s, though slight reductions in accuracy were observed, particularly in peak flow predictions. Balancing accuracy and efficiency is especially crucial in highly urbanized environments, and further research is needed to understand how these findings generalize to different watershed sizes and topographies.

Impact of Mesh Resolution: The mesh resolution used in this study was adequate for simulating urban flood dynamics, but further research is needed to assess the impact of finer resolutions on both accuracy and computational load. A finer mesh could improve accuracy in specific areas (e.g., narrow alleys), but at a significant computational cost. Establishing the optimal mesh resolution based on watershed characteristics and flood scenarios could enhance the model's overall effectiveness.

Future Research Directions: Future studies should focus on developing adaptive synchronization techniques that dynamically adjust based on changing watershed characteristics. Expanding the model to address other climate-related challenges, such as coastal storm surges and riverine floods, would further broaden its applicability.

In conclusion, the HC-SURF model has been confirmed as a reliable and efficient tool for urban flood simulation. With continuous improvements in mesh resolution, synchronization techniques, and real-world data integration, its utility in various urban environments will increase further.

## 7. Conclusions

This study aimed to verify the accuracy of the HC-SURF model and assess the effectiveness of the fixed-time synchronization technique. The model, which integrates 1D sewer flow and 2D surface water flow for urban flood prediction, was validated using experimental data, and flood analysis was conducted with the fixed-time synchronization approach on an actual urban watershed. The results are as follows:

The accuracy verification using experimental data showed that the HC-SURF model exhibited high accuracy across all rainfall scenarios, with minimal mass balance errors

between different inflow elements (roofs, inlets, manholes, etc.). MAE and RMSE results for inflows and outflows demonstrated errors of less than 1% in most scenarios, confirming the robustness of the model.

In the application of the fixed-time synchronization method to the actual urban watershed, longer synchronization times resulted in increased errors in surcharge and discharge volumes, but the errors were still within acceptable limits for large-scale flood prediction. The computational efficiency improved as the synchronization time increased up to 180 s, after which no significant further improvements were observed.

In conclusion, the HC-SURF model has been confirmed as a useful tool for urban flood management and real-time forecasting, providing high accuracy and computational efficiency. However, further research is needed to assess the sensitivity of synchronization times based on different watershed characteristics.

**Author Contributions:** Conceptualization, S.-B.S. and H.-J.K.; methodology, S.-B.S. and H.-J.K.; software, S.-B.S.; validation, S.-B.S. and H.-J.K.; formal analysis, S.-B.S.; investigation, S.-B.S.; resources, S.-B.S.; data curation, H.-J.K.; writing—original draft preparation, S.-B.S. and H.-J.K.; writing—review and editing, S.-B.S. and H.-J.K.; visualization, S.-B.S.; supervision, H.-J.K.; project administration, H.-J.K.; funding acquisition, H.-J.K. All authors have read and agreed to the published version of the manuscript.

**Funding:** This work was supported by the Korea Environment Industry & Technology Institute (KEITI) through the R&D Program for Innovative Flood Protection Technologies against Climate Crisis Project, funded by the Korea Ministry of Environment (MOE) (2022003470001).

**Data Availability Statement:** The datasets used and/or analyzed during the current study are available from the corresponding author on reasonable request.

**Acknowledgments:** The authors express their sincere gratitude to the Korea Environment Industry & Technology Institute (KEITI) and the Korea Ministry of Environment (MOE) for their funding and support through the R&D Program for Innovative Flood Protection Technologies against Climate Crisis Project. The authors would also like to thank their collaborators and team members for their valuable contributions to this research.

**Conflicts of Interest:** The authors declare no conflict of interest.

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
