# Peer review of "An Urban Flood Model Development Coupling the 1D and 2D Model with Fixed-Time Synchronization"

_water, doi:10.3390/w16192726_

Round 1
Reviewer 1 Report
Comments and Suggestions for Authors
The manuscript is within the scope of the journal and the content is meaningful. I think it can be published after minor revisions.
1. There is an error in the formula serial number, please correct it.
2. Please make Table 1 font smaller.
3. Fig. 4 puts the graph and table names on the same page.
4. Line: 242~245 Before and after is repeated.
5. Line: 275~283: Please place the introduction of the location of the study area in the first half of the article, and mark the location of Inlet et al.in detail.
6. Line: 385~392: Why is there such a trend ?
7. Table 4: Please make the standard format.
8. Please simplify your conclusion.
9. Please unify the reference format.
Author Response
Comments 1: There is an error in the formula serial number, please correct it.
Response 1 : Revised the number of formula corresponding to Section 2.3(Line 155 – 163)
Comments 2 : please make Table 1 font smaller.
Response 2: Table font size reduced by 1 (10 to 9, Table 1 – 5)
Comments 3 : Fig. 4 puts the graph and table names on the same page.
Response 3 : We knew that the legend in Figure 4 was not very visible, so We tried to correct it. To improve the visibility of the legend, We controlled the color and increased the font size (Figure 4)
Comments 4 : Line: 242~245 before and after is repeated.
Response 4 : corrected the overlapping sentences (Line 245-246).
Comments 5 : Line: 275~283: Please place the introduction of the location of the study area in the first half of the article, and mark the location of Inlet et al.in detail.
Response 5 : The composition of this paper consists of verifying the accuracy of the HC-SURF model through the application of a laboratory set and reviewing the effectiveness of the fixed-time synchronization method using the actual watershed. Thus, Section 3 describes the laboratory set and the accuracy test results of the HC-SURF model, and Section 4-5 describes the actual urban watershed and reviews the effectiveness of the fixed-time synchronization method.
Response 5 : Figure 9 has been modified to appear suitable for entrance, land use, soil, and hydrology. Information about the sewer network has been added to the text (lines 293-294)
Comments 6: Line: 385~392: Why is there such a trend?
Response 6: As the synchronization time increases, it is difficult to reflect the change in the water level with the linear interpolation method. There is a difference in the flow rate exchanged accordingly. In Figure 13, Discharge represents the capacity of the surface water flowing into the manhole and shows a tendency to decrease as the synchronization time increases. This is considered to be a problem caused by the limitation of the linear interpolation method for the change in water level before and after synchronization, although the fluctuation of the water level actually shows a steeper up-and-down curve as the synchronization time increases.
Comments 7 : Table 4: Please make the standard format.
Response 7 : We have modified the table according to standard format.
Comments 8 : Please simplify your conclusion.
The conclusions have been revised based on the reviewer's comments. In addition, the 'Discussion' section has been revised to write about the limitations of this study and the future required research
Comments 9 : Please unify the reference format.
Response 9 : We found the wrong part in the reference form and corrected it.
Reviewer 2 Report
Comments and Suggestions for Authors
The authors systematically address an important research topic – modeling urban floods. Their paper is well organized and for the most part clearly written. The mathematical development of the model is sketched out concisely but with enough detail to make it understandable to an interested reader. Their novel contribution of coupling a 1-D storm sewer model with a 2-D surface water model is shown in Figure 1, and detailed consideration of how the time stepping is implemented is given in Sections 4.3 and 5. They assess the validity of the model at the small scale by comparing it to a rainfall and urban drainage simulation (Section 3). They examine the feasibility of applying their model to an urban area (~ 5 km2) in Section 4, and suggest it reproduces the inundation induced by a heavy rainfall reasonably well. I think this research, if published, is likely to be widely cited as a contribution to urban flood modeling.
In Figure 2, all manholes are labeled “MH1”, whereas they should be MH1, MH2,… etc.
Captions for Figures 4 & 5 refer to “Grate” but “grate” does not occur in the text. Does this refer to the transverse inlet grid (line 184)? Clarify.
In the text, Figure 3 is said to address the model mesh (line 192). The figure does not show the mesh but only the elevations of the model components. There are 75,000 elements to represent the small drainage simulation… how do the authors determine the appropriate number for the coupled models? This question becomes especially important to address in the case of the larger modeled area (Sillim district). To what extent could their results for synchronization time (Section 5) depend on the grid used? Would different gridding choices lead to better/worse results? See line 337.
The color key for Figures 4 & 5 is difficult to see. I suggest using different colors or fills.
Discussion of Figures 6 & 7 fails to note or explain an easily visible systematic effect: for all inlets and roofs the model has a delayed rise and a slower decline than observed. This is easily seen in the charts for Inlet 4 and Roof 4. How does this model artifact arise? Is it potentially significant in actual applications (Section 4)? If some model flow is systematically delayed, the model could under-estimate peak flow and inundation.
The different components of Figure 9 and their legends are highly detailed. If it is important to show these, then they could be made larger. For example, place 9a above 9b; this would let each part become larger and more readable.
Similar to the previous bullet, it is really difficult to for me to actually compare what is blue and what is gray in Figure 10. Does blue lie on top of gray? Vice versa? Perhaps a scheme that emphasizes only the areas where the model and trace map disagree would work better (e.g., gray, not blue = red; blue, not gray = purple). Similar issues occur in Figure 14.
The text (lines 336-338) suggest that Figure 11 will address the size of the computational time. But the figure itself is purely schematic; is it actually needed? I think the point might be clear without it.
Line 391: 10 m3/sec, not 10 m/sec.
Line 445: I don’t think I understand… are you suggesting that 120-sec synchronization may be most efficient in all cases?
I think Section 6 would be better written in the form of a Discussion rather than as Conclusions. For example, at the end of Section 5 it is mentioned that “further research is required” and the factors that need to be considered to find an “appropriate synchronization time” are mentioned. This might lead to discussion of how this paper will motivate other researchers and the considerations that might apply in extending the model to other climates, other topographies, other city plans.
Author Response
Comments 1 : In Figure 2, all manholes are labeled “MH1”, whereas they should be MH1, MH2,… etc.
Response 1 : Manhole and Inlet 5 modification in Figure 2(Inlet 5 -> Grate).
Comments 2 : Captions for Figures 4 & 5 refer to “Grate” but “grate” does not occur in the text. Does this refer to the transverse inlet grid (line 184)? Clarify.
Response 2 : Revised Inlet5 in Figure 2 to grate(Table 1, Line 184).
Comments 3 : In the text, Figure 3 is said to address the model mesh (line 192). The figure does not show the mesh but only the elevations of the model components. There are 75,000 elements to represent the small drainage simulation… how do the authors determine the appropriate number for the coupled models? This question becomes especially important to address in the case of the larger modeled area (Sillim district). To what extent could their results for synchronization time (Section 5) depend on the grid used? Would different gridding choices lead to better/worse results? See line 337.
Response 3 : Figure 3 has been modified to represent the grid.
The laboratory area used an unstructured triangular mesh, with an average element size of 0.05 m. As a result, the average area of each element is 0.00125 m², and the total area is 93.75 m², comprising 75,000 elements in total. The DEM provided in the laboratory data has a high-resolution accuracy of 0.01 m. However, if the average element size is set to 0.01 m, the total number of elements increases to 375,000, significantly lengthening the computation time. Additionally, in reference [28], there was a case where a road mesh with an average element size of 0.05 m produced results similar to the observed data. In reference [27], a comparison was made between high-resolution mesh simulations with an element size of 0.01 m for roof grids and the nonlinear storage method using the SWMM's RUNOFF module, which yielded similar analysis results. Therefore, the overall mesh size for the experimental basin was set to 0.05 m. For the actual basin, Sillim Basin, the DEM data resolution is 1 m, and based on this, an unstructured triangular mesh with a minimum element size of 1 m² was created, comprising a total of 67,195 elements.
As you mentioned, the synchronization time (Section 5) is unlikely to be significantly affected by the mesh resolution. Synchronization is performed by comparing the water level of the grid where the manhole is located with the water level of the manhole, and since the surface water level is the same within the grid, the mesh resolution does not play a critical role in this process. If the mesh resolution were finer than the catchment area of the manhole, there could be differences in synchronization time. However, this is likely infeasible for urban basin-scale simulations. We believe that further research on this matter is necessary and will add this discussion to the relevant section of the paper.
Comments 4 : The color key for Figures 4 & 5 is difficult to see. I suggest using different colors or fills.
Response 4 : Increased visibility by changing to a more distinct color (Fig4 & 5)
Comments 5 : Discussion of Figures 6 & 7 fails to note or explain an easily visible systematic effect: for all inlets and roofs the model has a delayed rise and a slower decline than observed. This is easily seen in the charts for Inlet 4 and Roof 4. How does this model artifact arise? Is it potentially significant in actual applications (Section 4)? If some model flow is systematically delayed, the model could under-estimate peak flow and inundation.
Response 5 : First, we performed accuracy verification with a large set of laboratories to evaluate the accuracy of HC-SURF models and perform 2D synchronization for each time step (dt). Therefore, as pointed out, we want to clarify that the fixed time synchronization in Section 4 is not affected.
In Figures 6 and 7, a delay rise was observed at Inlet 4 and ROOP4, while the flow curves of the other Inlet and ROOPs were found to be in relatively good agreement with the observed data. The cause of the delay rise observed at Inlet 4 and ROOP4 can be attributed to a variety of factors, such as the slope of the mesh, the roughness coefficient used, or the discretization of the numerical model. However, according to Reference 29, this is likely due to a small groove or protrusion in the laboratory setup. As suggested, we added a description of Figures 6 and 7
Comments 6 : The different components of Figure 9 and their legends are highly detailed. If it is important to show these, then they could be made larger. For example, place 9a above 9b; this would let each part become larger and more readable.
Response 6 : Figure 9 shows the topographic information of the numerical grid and sewer network configuration, which was not considered an important factor. Therefore, we removed 'soil' from the original picture and enlarged the image to improve visibility.
The name of Figure 9 was changed to "Hydraulic-Hydrological Data Analysis and Initial Model Setting".Comments 7 : Similar to the previous bullet, it is really difficult to for me to actually compare what is blue and what is gray in Figure 10. Does blue lie on top of gray? Vice versa? Perhaps a scheme that emphasizes only the areas where the model and trace map disagree would work better (e.g., gray, not blue = red; blue, not gray = purple). Similar issues occur in Figure 14.
Response 7 : Since the comparison picture between the inundation trace and the simulation result is hard to see, we increased visibility by dividing the part where the simulation result fits and doesn't fit as you suggested.
Comments 8 : The text (lines 336-338) suggest that Figure 11 will address the size of the computational time. But the figure itself is purely schematic; is it actually needed? I think the point might be clear without it.
Response 8 : Fig 11 has been deleted
Comments 9 : Line 391: 10 m3/sec, not 10 m/sec.
Response 9 : I modified it(Line 393).
Comments 10 : Line 445: I don’t think I understand… are you suggesting that 120-sec synchronization may be most efficient in all cases?
Response 10 : A decreasing trend was observed up to 180 s, but it was incorrectly entered as 120 s. When analyzing the Sillim basin, the computational efficiency was found to continuously decrease by 180 s. Besides this, increasing the synchronization time only reduced the number of synchronization events, resulting in a slight improvement in the computational efficiency. Therefore, simplification up to 180 s was considered effective from a computational efficiency perspective. Considering this information, some modifications were made (line 445-446).
Comments 11 : I think Section 6 would be better written in the form of a Discussion rather than as Conclusions. For example, at the end of Section 5 it is mentioned that “further research is required” and the factors that need to be considered to find an “appropriate synchronization time” are mentioned. This might lead to discussion of how this paper will motivate other researchers and the considerations that might apply in extending the model to other climates, other topographies, other city plans.
Response 11 : At the end of Section 5, the description of the factors needed to determine 'further research' and 'appropriate synchronization time' was extended to a more detailed discussion.
Reviewer 3 Report
Comments and Suggestions for Authors
Line 18: please clarify why ‘ as the sync time increases, the errors in surcharge and discharge also increases’ ?
Line 76: need to add citation for SWMM 5.2
Line 78: need to add citation for the present researchers
Line 154 and 158: why do the calculations for discharge and surcharge used weir and orifice formulas, correspondingly?
Line 204: need to clarify the T1, T2, and T3 rainfall characteristics for their first occurrence
After the results analysis section, could you add a discussion section?
How does your dual-coupled model outperform other existing 1d-2d coupled model?
Please discuss the limitations and future work of your research
Author Response
Comments 1 : Line 18: please clarify why ‘ as the sync time increases, the errors in surcharge and discharge also increases’ ?
Response 1 : Longer synchronization times resulted in increased surcharge and discharge errors as water level changes could not be accurately reflected (Line 18-20). The content has been corrected accordingly.
Comments 2 : Line 76: need to add citation for SWMM 5.2
Response 2 : Added citation SWMM 5.2 (Line 77)
Comments 3 : Line 78: need to add citation for the present researchers
Response 3 : Regarding the above phrase, we added references (line 78-80)
Comments 4 : Line 154 and 158: why do the calculations for discharge and surcharge used weir and orifice formulas, correspondingly?
Response 4 : Recent work has commonly used orifice and beam formulations to simulate surface water-manhole flow exchange. While there are other exchange methods, this work has adopted a widely used formulation for consistency.
Comments 5 : Line 204: need to clarify the T1, T2, and T3 rainfall characteristics for their first occurrence
Response 5 : We have added explanations for T1, T2, and T3.
Comments 6 : After the results analysis section, could you add a discussion section?
Response 6 : The conclusions have been revised based on the reviewer's comments. In addition, the 'Discussion' section has been revised to write about the limitations of this study and the future required research.
Comments 7 : How does your dual-coupled model outperform other existing 1d-2d coupled model?
Response 7 : Conventional 1D-2D coupled models are often specific to individual goals, with their combining methods and source exchange techniques varying depending on the purpose (speed or accuracy) of the model. On the other hand, the combined model proposed in this study performs flow synchronization at fixed time intervals, enabling fast simulation while maintaining a certain level of accuracy compared to flow exchange methods based on 2D time steps. Thus, this approach enables both fast and accurate simulations.
However, it may be suitable for extensive simulations in urban watershed units, but may not be suitable for high-precision analysis. Therefore, it is necessary to conduct further studies on various watersheds as described in the discussion session.
Comments 8 : Please discuss the limitations and future work of your research
Response 8 : This study verified the accuracy of the HC-SURF model using laboratory data and tested the effectiveness of the fixed-time synchronization method on the actual watershed. The watershed used for the verification was an area with high building density, and it was difficult to capture the water level fluctuation as the synchronization time increased due to the narrow space between buildings. In the future, we plan to evaluate the accuracy and speed of various synchronization times under various watershed conditions and seek the application strategy of the synchronization technique according to the watershed characteristics. This was added to the discussion section.
Round 2
Reviewer 3 Report
Comments and Suggestions for Authors
The revised version looks good to me.